# Epidemiological Study of p16 Incidence in Head and Neck Squamous Cell Carcinoma 2005–2015 in a Representative Northern European Population

**DOI:** 10.3390/cancers14225717

**Published:** 2022-11-21

**Authors:** Mari Mylly, Linda Nissi, Teemu Huusko, Johannes Routila, Samuli Vaittinen, Heikki Irjala, Ilmo Leivo, Sami Ventelä

**Affiliations:** 1Department for Otorhinolaryngology, Head and Neck Surgery, University of Turku and Turku University Hospital, Kiinamyllynkatu 4-8, 20521 Turku, Finland; 2Department of Oncology, University of Turku and Turku University Hospital, Kiinamyllynkatu 4-8, 20521 Turku, Finland; 3Department of Pathology, Turku University Hospital and Turku University, Kiinamyllynkatu 10, 20521 Turku, Finland; 4Biomedical Institute, University of Turku, Kiinamyllynkatu 10, 20520 Turku, Finland; 5Turku Bioscience Centre, University of Turku and Åbo Akademi University, Tykistökatu 6, 20520 Turku, Finland

**Keywords:** HNSCC, OPSCC, HPV, p16, epidemiology, incidence, survival

## Abstract

**Simple Summary:**

Human papillomavirus (HPV) infection is a known risk factor for developing head and neck squamous cell carcinoma (HNSCC), especially in the oropharyngeal region. Determining HPV status is important, as both survival and treatment response are dependent on HPV expression. p16 immunohistochemistry is a widely used method to detect HPV positivity in cancer. The aim of this study was to determine the prevalence and associations of p16 positivity with survival in all patients diagnosed with new HNSCC in Southwest Finland between 2005 and 2015. p16 positivity involved a high locoregional correlation, as 72% of all p16-positive tumors were located in the oropharynx. p16 positivity in the oropharynx was associated with an increased likelihood of lymph node metastasis, a smaller primary tumor size and a sparse history of smoking and alcohol consumption. Notably, in all studied categories, HNSCC patients benefited from combining chemotherapy with radiotherapy, regardless of p16 expression.

**Abstract:**

The incidence of human papillomavirus (HPV)-associated head and neck squamous cell carcinomas (HNSCC) has increased globally. Our research goal was to study HNSCC incidence in a representative Northern European population and evaluate the utility of the HPV surrogate marker p16 in clinical decision-making. All new HNSCC patients diagnosed and treated in Southwest Finland from 2005–2015 (*n* = 1033) were identified and analyzed. During the follow-up period, the incidence of oropharyngeal (OPSCC) and oral cavity squamous cell carcinoma (OSCC) increased, while the incidence of laryngeal squamous cell carcinoma (LSCC) decreased. This clinical cohort was used to generate a population-validated tissue microarray (PV-TMA) archive for p16 analyses. The incidence of p16 positivity in HNSCC and OPSCC increased in southwest Finland between 2005 and 2015. p16 positivity was mainly found in the oropharynx and was a significant factor for improved survival. p16-positive OPSCC patients had a better prognosis, regardless of treatment modality. All HNSCC patients benefited from a combination of chemotherapy and radiotherapy, regardless of p16 expression. Our study reaffirms that p16 expression offers a prognostic biomarker in OPSCC and could potentially be used in cancer treatment stratification. Focusing on p16 testing for only OPSCC might be the most cost-effective approach in clinical practice.

## 1. Introduction

Head and neck squamous cell carcinomas (HNSCC) contain a heterogeneous group of tumors located in the oral cavity, larynx, and pharynx [1]. Tobacco and alcohol consumption are known risk factors for HNSCC, but human papillomavirus (HPV) infection, especially type 16, has been established to have a causal role in the progression of HNSCC, particularly in the oropharynx [2,3]. Moreover, various epidemiologic studies have indicated that oropharyngeal cancers are increasingly associated with HPV infection, as the incidence of non-HPV-related oropharyngeal squamous cell carcinoma (OPSCC) has decreased [4,5,6,7]. While this change in incidence is partially due to reduced tobacco consumption, it is also linked to increased awareness and identification of HPV [1,4].

HPV is a sexually transferred virus, and the latency between oral exposure and oropharyngeal cancer has been determined to be between 10 and 30 years [8]. Carcinogenesis of HPV infection is caused by multiple cell mutations after high-risk HPV DNA is integrated into the host genome. The host cell produces viral oncoproteins E6 and E7 that initially degrade the tumor suppressors p53 and retinoblastoma protein (Rb), causing uncontrolled cell cycle progression. p16 is a tumor suppressor protein that prevents cell cycle progression by inhibiting the activation of Rb. Thus, in HPV-driven HNSCCs, p16 protein is overexpressed, as the E7 oncoprotein causes degradation of Rb. [7,9,10,11,12] On the other hand, the p16 gene is inactivated and its protein production limited in non-HPV-related HNSCC [13]. Therefore, p16 positivity is a widely used surrogate marker for HPV-positive cancers, as various studies have indicated a high correlation between the detection of HPV and p16 overexpression by immunostaining [14,15]. Additionally, a major benefit of immunohistochemistry for p16 is that it has a high sensitivity for HPV and a low cost [7,16,17,18].

Dividing OPSCC patients into two groups by HPV expression is useful, as HPV-positive and HPV-negative patients differ in characteristics: patients with HPV-positive tumors tend to be younger, healthier, and smoke less. Furthermore, HPV-positive OPSCC patients are typically males and have a higher average number of sexual partners than HPV-negative OPSCC patients [6,10,19]. Generally, cancer treatments are also more effective for HPV-positive cancers [20,21,22]. As socioeconomic status, age and especially molecular status differ between HPV-positive and HPV-negative cancers, the overall survival is also different in these two groups, favoring patients with an HPV-related disease [21,23,24].

Less intensive cancer treatments could potentially accomplish similar efficacy in HPV-positive OPSCC, with less toxicity and better quality of life [25,26,27]. However, there are no validated methods to identify the ideal group of patients for less intensive treatments from those in need of intensive multimodality therapies [27]. Furthermore, as premalignant lesions for HPV-positive OPSCCs are unknown, the utilization of screening is not possible [28]. Many biomarkers have been studied to identify patients in clinical practice with the best prognosis, but the results have been inconclusive [29,30,31].

The main purpose of this study was to examine the associations between p16 expression and overall survival (OS) among HNSCC patients in a representative and previously verified population-validated setting [32]. Additionally, information on alcohol and tobacco consumption along with other significant factors, such as the TNM criteria or primary tumor site, allows the evaluation of potential confounders in an analytical framework. Inclusion bias caused by health insurance or socioeconomic status-related issues is avoided, as all Finland’s HNSCC patients in need of oncological treatment are treated in tertiary referral centers according to national treatment guidelines [32].

## 2. Materials and Methods

### 2.1. Participants and Data Collection

In this study, a population-based cohort was composed of all new HNSCC patients who were treated between 2005 and 2015 in the Southwest Finland tertiary referral center of Turku University Hospital (TUH). The study protocol is shown in Figure 1 and in previous studies [32,33]. TNM criteria were used to stage tumors at the time of diagnosis. Treatment protocols were collected from meetings of the multidisciplinary Tumor Board for head and neck cancer. Overall survival was defined from end-of-treatment to end-of-follow up or death. Survival status was gathered from medical records of TUH, which is connected to the Finnish National Population Information System database. Patients’ end-of-follow-up dates were recorded from the time of last data in the TUH records. Information on alcohol and tobacco use was collected from the patient records. Cumulative use of tobacco was estimated as pack years (PY), whereby the amount of daily smoked packs was multiplied by the period of tobacco use in years.

The study and utilization of human tissue samples were approved by the Finnish national authority for medicolegal affairs (V/39706/2019), regional ethics committee of University of Turku (51/1803/2017) and Auria Biobank scientific board (AB19-6863). Formalin-fixed and paraffin-embedded (FFPE) tumor samples were obtained from the pathology archives of Auria Biobank. Final tissue microarray (TMA) blocks of duplicate 0.6 mm cores were made in TMA Grand Master (3D Histech). Normal liver samples were included in each block for orientation. The authors affirm that the study was conducted following the rules of the Declaration of Helsinki of 1975, revised in 2013.

### 2.2. Immunohistochemistry

FFPE blocks were cut into 6 µm sections. Immunohistochemical staining of p16 (Roche/Ventana clone E6H4) was performed in a Ventana Bench-Mark XT staining automate (Ventana Medical Systems, Inc., Oro Valley, AZ, USA) in the laboratory of clinical pathology. Two independent authors (M.M. or J.R. and I.L.) analyzed the immunohistochemical staining, and differences were conferred until consensus was reached. p16 immunostaining was graded positive if at least 70% of cells displayed strong nuclear and cytoplasmic staining intensity.

### 2.3. Statistical Analysis

Statistical analyses were conducted using SPSS 27 software (SPSS, IBM, 1 New Orchard Road Armonk, New York 10504-1722, NY, USA). Overall survival and disease-free survival curves were plotted using the Kaplan-Meier method. The log rank test was run to determine the significance of differences in survival distribution between patient groups. Binomial logistic regression was performed to ascertain the effects of age, sex, smoking, alcohol consumption, tumor site, TNM class, stage, and treatment on the likelihood of patients being included in the TMA. Two-sample t tests and chi-square tests were used to evaluate differences between p16-positive and p16-negative OPSCC patients. The multivariable Cox proportional hazards method was applied to adjust the survival effect of p16 and treatment type on age, smoking, alcohol use, T-class, and nodal positivity. Hazard ratios (HR) with 95% confidence intervals (CI) and *p* values were reported. *p* values of less than 0.05 were considered significant.

## 3. Results

### 3.1. Epidemiology of HNSCC in Southwest Finland 2005–2015

The population of Southwest Finland increased from 683,000 to 697,000 during 2005–2015 [34]. The HNSCC patient cohort for epidemiological, clinical, and p16 evaluations was formed by identifying and including all patients diagnosed and treated for new HNSCC in the background population cohort within the years 2005–2015 (*n* = 1033, Figure 1). The annual HNSCC incidence rates per 100,000 population varied between 10.05 (2008) and 16.72 (2013; Figure 2). The absolute number of newly diagnosed HNSCC cases in 2015 (*n* = 109) was 34.6% higher than that in 2005 (*n* = 81). Among 1033 HNSCC patients, the median age of newly diagnosed HNSCC was 65 (range 23–95). Descriptive statistics of patient characteristics are shown in Table 1. In addition, the survival effect of selected variables was calculated and reported based on a prognostic model previously described by Denissoff et al. [33]. The mean and median follow-up times were 38.6 and 49.3 months, respectively. The five-year overall survival (OS), disease-specific survival (DSS), and disease-free survival (DFS) were 55.1%, 69.4%, and 60.6%, respectively, as presented in Appendix A. Lymph node metastases were detected in 38.2% (*n* = 395) of HNSCC patients. Regarding cancer therapy, the majority of HNSCC patients received either surgery only (35%) or combined treatment, including surgery and radiotherapy (RT) or surgery and chemoradiotherapy (CRT; 36%). The remaining HNSCC patients received either definitive RT (8.2%), CRT (15.4%), or palliative therapy (5.3%). Patients who received RT or CRT either as definitive or adjuvant treatment were included for further analysis along with patients who were treated by surgery only. Patients in the RT group had worse OS than patients treated by CRT (HR 0.57; 95% CI 0.41–0.81; *p* value 0.002) and surgery only (HR 0.63; 95% CI 0.43–0.93; *p* value 0.019). The survival effect was analyzed using a multivariable Cox proportional hazards model adjusting for T-class, nodal positivity, consumption of alcohol and tobacco, and patient age.

### 3.2. Locoregional Distribution of HNSCC between 2005 and 2015

The most common tumor site was the oral cavity (*n* = 505), followed by cancers of the oropharynx (*n* = 193), as shown in Table 1. The absolute number of patients diagnosed with squamous cell carcinoma of the oral cavity (OSCC) in 2015 (*n* = 59) was 28.3% higher than that in 2005 (*n* = 46). Furthermore, the number of oropharyngeal SCC (OPSCC) patients rose by 100.0% during the same period (*n* = 11 in 2005 and *n* = 22 in 2015). Meanwhile, the number of patients with a new SCC of the larynx (LSCC) was 42.1% lower in 2015 (*n* = 11) than in 2005 (*n* = 19).

The existence of known risk factors, such as age, sex, and consumption of tobacco and alcohol, was also analyzed. The site-specific presence of risk factors is presented in Table 2. A heavy smoking history of twenty or more pack years (PY) was most common in patients with LSCC (83.2%, *n* = 153/184) and hypopharyngeal cancer (HPSCC, 72.5%, *n* = 29/40). Additionally, current use of alcohol was most common in patients with HPSCC (37.5%, *n* = 15/40). The association with male sex was notable in LSCC and OPSCC, with 86.4% (*n* = 159/184) and 77.8% (*n* = 150/193) of patients being males, respectively. Patients with SCC of the oropharynx were remarkably younger by the time of diagnosis compared to other locations, with only 31.6% (*n* = 61/193) of OPSCC patients being 65 years or older. On the other hand, 61.8% (*n* = 312/505) and 60.0% (*n* = 24/40) of patients with SCC of the oral cavity and hypopharynx, respectively, were over 65 years old at the time of HNSCC diagnosis.

### 3.3. Epidemiology of Oropharyngeal Squamous Cell Carcinoma (OPSCC)

Our epidemiological site-specific and locoregional analyses showed that the highest increase in new HNSCCs from 2005 to 2015 existed in OPSCC. Our HNSCC cohort included 193 OPSCC patients, of whom 77.8% (*n* = 150) were males. The median age of OPSCC patients was 60 years, and 63.2% (*n* = 122) had a smoking history of at least 20 PY. Moreover, 47.7% (*n* = 92) of the patients were previous alcohol consumers, and 32.1% (*n* = 62) were current consumers. The incidence of OPSCC varied from 1.46 (2007) to 3.74 (2014) per 100,000 population. The annual OPSCC incidence during 2005–2015 is illustrated in Figure 3.

In the OPSCC cancer treatment algorithm, radio-RT and chemoradiotherapy (CRT) are the main treatment options. Of 193 OPSCC patients, 7.8% (*n* = 15) received definitive radiotherapy, and 23.3% (*n* = 45) received definitive chemoradiotherapy as treatment. Most patients (52.8%, *n* = 102) were treated with a combination of surgery and CRT, while a minority (3.6%, *n* = 7) of patients received a combination of surgery and RT. In addition, the proportion of patients treated by surgery only was 3.6% (*n* = 7). Palliative care or no treatment at all was offered to 8.8% (*n* = 17) of patients.

The five-year OS, DSS, and DFS of patients with OPSCC were 54.4%, 66.3%, and 61.7%, respectively. Treatment outcomes were compared between CRT (*n* = 147) and RT (*n* = 22) treated patients. Patients who received RT or CRT as an adjuvant treatment were included along with patients who received definitive RT or CRT. Patients treated with CRT had better OS (*p* < 0.001), DSS (*p* = 0.002), and DFS (*p* = 0.008), as shown in Figure 4. OS was 63.9% in the CRT group and 31.8% in the RT group, DSS was 75.5% in the CRT group and 50.0% in the RT group, and DFS was 71.4% in the CRT group and 50.0% in the RT group. The group of patients treated by surgery only was not analyzed.

### 3.4. Establishment of a Population-Validated Tissue Microarray (PV-TMA) Corresponding to an Epidemiological HNSCC Background Population

Of 1033 patients, 685 (66.3%) had a tumor sample available for PV-TMA, as shown in Figure 1. Clinical data of TMA patients were compared to the background population of all HNSCC patients treated in the Southwest Finland region from 2005–2015. The representativeness of the TMA cohort against the background HNSCC population is presented in Table 3. The established TMA was confirmed to be representative in terms of age, sex, alcohol and tobacco consumption, T-class, and stage, while an uneven distribution of N-class was observed. However, the difference in patients presenting with nodal metastasis was rather modest (38% in background population vs. 43% in PV-TMA). Most importantly, there was no statistically significant difference in OS (*p* = 0.200), DSS (*p* = 0.146), or DFS (*p* = 0.125) between the PV-TMA and background HNSCC populations (Figure 5). Thus, the established PV-TMA can be considered to represent HNSCC patients treated in the Southwest Finland region from 2005–2015.

### 3.5. P16 Immunohistochemical Analyses

In p16 immunohistochemical staining, 593 of 685 patients in the PV-TMA cohort had analyzable p16 staining. Representative examples of immunohistochemical staining are presented in Figure 6.

A total of 493 patients (83.1%) had p16-negative disease, while 100 patients (16.9%) had p16-positive disease. Of these 100 p16-positive diseases, 72 originated in the oropharynx. Table 4 depicts the p16 results by primary tumor site. Notably, there was very low p16 positivity in non-OPSCC regions.

Regarding OPSCC patients, 52.9% (*n* = 72) were p16-positive, and 47.1% (*n* = 64) were p16-negative. In terms of sex, 55.2% of males and 45.2% of females were p16-positive. Furthermore, 80.6% of all p16-positive OPSCC patients were males. The median age of p16-positive patients was 60 years, while the median age of p16-negative patients was slightly higher, 62.5 years at the time of diagnosis. However, the difference was not statistically significant (*p* = 0.125) in this subgroup of the cohort. p16-positive diseases were more likely to spread to lymph nodes than their p16-negative counterparts (*p* = 0.032). However, T classification was lower in p16-positive primary tumors (*p* = 0.031).

In p16-positive OPSCC patients, smoking history was remarkably rare (*p* < 0.0001), as only 40.3% (*n* = 29/72) had a smoking history of 20 PY or more in comparison to 85.9% (*n* = 55/64) of p16-negative patients. In addition, p16-positive patients used less alcohol, with 19.4% (*n* = 14/72) being current and 29.2% (*n* = 21/72) being former consumers of alcohol. However, 45.3% (*n* = 29/64) of p16-negative patients had current and 65.6% (*n* = 42/64) had previous alcohol consumption. *p* values reached <0.001 in terms of both current and previous alcohol consumption. The incidence of p16-positive OPSCCs varied between 0.29 (2007) and 1.88 (2011) per 100,000 population (Figure 7). The absolute number of newly diagnosed annual p16-positive OPSCCs rose from 11 patients in 2005 to 22 patients in 2015.

p16 positivity in OPSCC was associated with better OS (*p* < 0.001), DSS (*p* < 0.001), and DFS (*p* < 0.001), as shown in Figure 8. In other SCCs, excluding tumors of the oropharynx, p16 positivity did not correlate with better OS (*p* = 0.264) or DSS (*p* = 0.095). However, p16-positive patients with tumors in sites other than the oropharynx had better DFS (*p* = 0.031), as illustrated in Appendix A. Comparing RT to CRT treatments, both p16-positive and p16-negative patients benefited from combining chemotherapy with RT, as OS was better in the CRT cohort vs. the RT group in both p16-positive (71.1% vs. 57.1%) and p16-negative (46.2% vs. 7.7%) patients (Appendix A). However, the difference reached statistical significance only in p16-negative OPSCC (*p* = 0.009 in p16-negative and *p* = 0.237 in p16-positive patients).

To conclude, p16-positive patients had better OS, regardless of treatment modality (HR 0.64; 95% CI 0.43–0.95; *p* value 0.028). The survival effect was analyzed using a multivariable Cox proportional hazards model adjusting for T-class, nodal positivity, consumption of alcohol and tobacco, patient age, and treatment type.

## 4. Discussion

This extensive population-based cohort study demonstrates the associations between HPV surrogate marker p16 expression and HNSCC in a representative Northern European population of 700,000 people. The incidence of HNSCC, OPSCC, and p16 positivity also increased in this region, consistent with the results of other recent studies [4,6,35]. Importantly, p16 immunohistochemistry was confirmed to provide a useful tool in the identification of HNSCC patients with a better prognosis and better treatment response. As we had over 1000 patients with a new HNSCC in an 11-year follow-up, our findings are extremely reliable and translatable directly to clinical practice.

Information about known risk factors, such as tobacco and alcohol consumption, along with T-class and tumor stage were collected, as these factors are associated with more aggressive HNSCC and poorer survival rates [2,21,24,33]. The indisputable impact on survival was also shown in this study, as each factor was associated with a worse prognosis. Epidemiologically, alcohol and tobacco are well-known risk factors for HNSCC, which was also evident in our HNSCC cohort. Twenty-four percent of HNSCC patients had moderate or excessive alcohol consumption, while the same is true for 13% of the Finnish working-age population [36]. Tobacco consumption was also remarkably common in the HNSCC cohort, as 68.2% of patients smoked daily at the time of diagnosis, whereas only 16.6% of the average Finnish working-age population smoke daily [37].

Above all, in this study, high p16 expression was confirmed to be a significant prognostic factor for 5-year overall survival and 5-year disease-free survival in HNSCC, OPSCC and even in palliative patients receiving no treatment at all. To specify the role of HPV more closely, we explored patient cohorts by different subsites. There was no statistically significant impact of p16 expression on overall survival in HNSCC when oropharyngeal cancers were excluded. However, this analysis was complicated by the fact that a very significant portion of HNSCC tumors occurring outside the oropharynx are p16 negative (83–95% depending on the location of the tumor; Table 4). Furthermore, our results showed a very strong locoregional expression profile of p16 (only 4–5% of OSCC, LSCC, and HPSCC were p16 positive); therefore, it would be most cost-effective to focus routine testing of p16 in clinical practice for only OPSCC.

The difficulty of choosing the optimal cancer treatment method for HNSCC is due to the lack of clinically validated biomarkers. Consequently, much expectation has been placed on HPV detection and p16 staining for OPSCC cancer treatment de-escalation. However, our results do not truly support the de-escalation strategy in p16-positive patients because, in all studied settings, the prognosis was significantly better in the CRT treatment group than in RT-treated patients. Therefore, our results strongly suggest that successful de-escalation of p16-positive OPSCC patients would likely require novel biomarkers alongside p16 to predict cancer sensitivity to radio- and/or chemoradiotherapy [31,38,39,40,41]. Furthermore, the strong locoregional specificity of p16 expression strongly supports the hypothesis that, in future studies, the location of HNSCC may play a significant role in the functionality of the biomarker [29,42,43].

Particularly interesting phenomena in future HNSCC epidemiology will be the results of the HPV vaccination. Recent studies have indicated that the prevalence of vaccine-type oral HPV (types 6, 11, 16, and 18) is significantly lower in vaccinated adults than in unvaccinated adults [44,45,46]. Our results advocate the importance of vaccinating males against HPV, as 80.6% of all our p16-positive OPSCC patients were males. The HPV vaccine was introduced in the Finnish national vaccination program in 2013 for girls and in 2020 for boys, and it is offered for 10- to 12-year-old citizens [47]. As the youngest p16-positive patient in our study was 36 years old, the effect of vaccination was not yet apparent in this study. In the future, it would be interesting to study whether the prevalence of HPV-positive oropharyngeal cancers will also be reduced in Finland, as indicated by various recent global studies [44,45,46].

The main strength of this study is its representative population-based patient collection, as it included all new HNSCC patients diagnosed and treated in the southwest Finland area within an 11-year period, covering approximately one-sixth of Finland’s HNSCC patients. Inclusion bias caused by health insurance or socioeconomic status-related issues was also avoided, as all HNSCC patients in need of oncological treatment are referred to tertiary referral centers according to the national treatment guidelines. Due to this arrangement, all patients are given the opportunity to receive the most beneficial treatment. Additionally, as all treatments are based on national treatment guidelines, a confounding factor caused by the variance in given treatments from different hospitals and individual clinicians is abrogated. In addition, a comprehensive electronic medical record system provides an effective patient follow-up regimen, allowing the study cohort to remain almost complete. Few patients were lost during the follow-up, for instance, due to migration. Thus, our cohort represents real-life patient material, which increases the usability of these results in clinical decision-making.

However, there were also limitations in this study. As mentioned above, the number of patients in the locoregional p16 expression analyses was relatively low when OPSCCs were excluded. Additionally, although treatment decisions are based on national treatment guidelines, patients’ health status is additionally acknowledged; a certain group of patients tolerate only radiotherapy, and thus, these patients do not receive chemotherapy at all. Thus, there are certain difficulties in comparing overall survival in different treatment groups, as the method of treatment was not randomized.

## 5. Conclusions

Our study demonstrated that the incidence of HNSCC, OPSCC, and p16 positivity increased during the years 2005–2015 in a representative northern European population of 700,000 people. p16 positivity was mainly encountered in the oropharynx; in this site, p16 proved to be a significant independent factor for improved survival. Additionally, known risk factors, such as tobacco and alcohol consumption, along with T-class and tumor stage, were indicated to have a strong impact on poorer survival rates. Moreover, epidemiologically, HNSCC patients used more tobacco and alcohol than the average population. Overall survival was better for p16-positive patients regardless of treatment modality. The use of p16 expression as a relevant biomarker for HPV-positive OPSCC was thus confirmed to be a remarkable tool to identify patients with better prognoses.

## Figures and Tables

**Figure 1 cancers-14-05717-f001:**
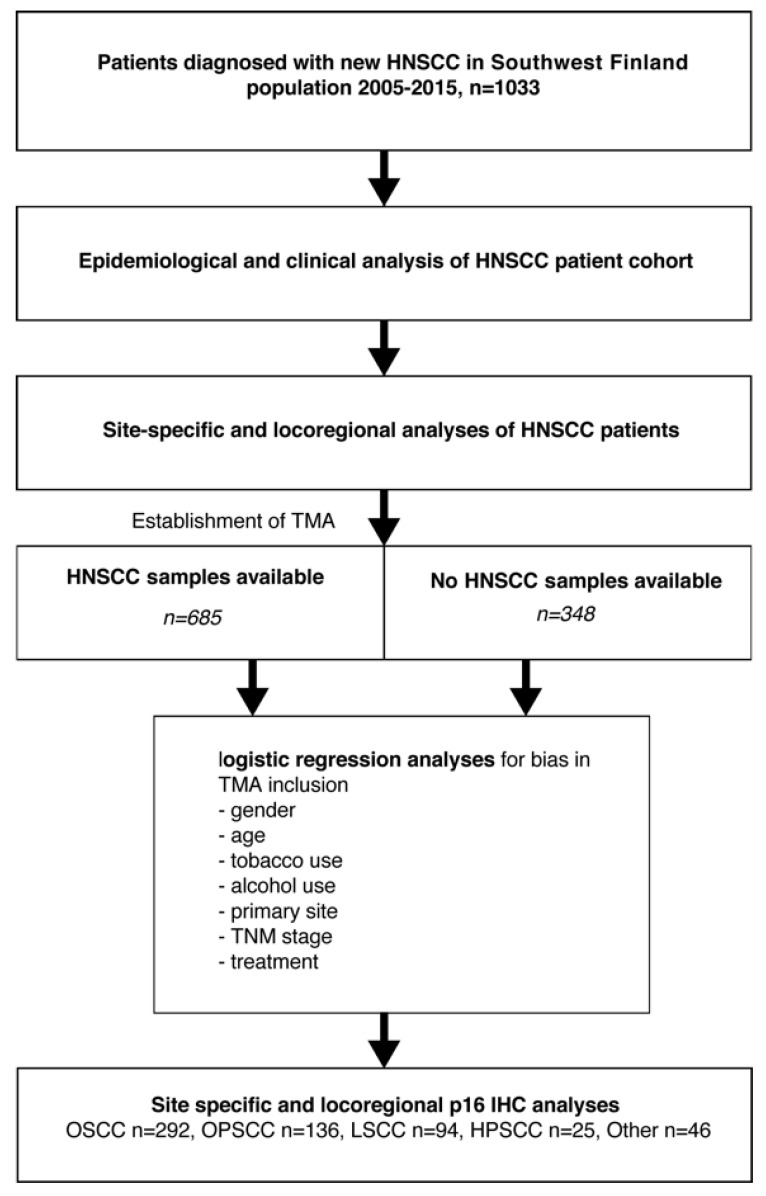
The study protocol for the identification and analysis of all patients with new HNSCC in the Southwestern Finland region from 2005–2015 (*n* = 1033). The same clinical cohort was used in the formation of an unbiased and population-validated HNSCC tissue microarray (TMA) archive for p16 immunostaining.

**Figure 2 cancers-14-05717-f002:**
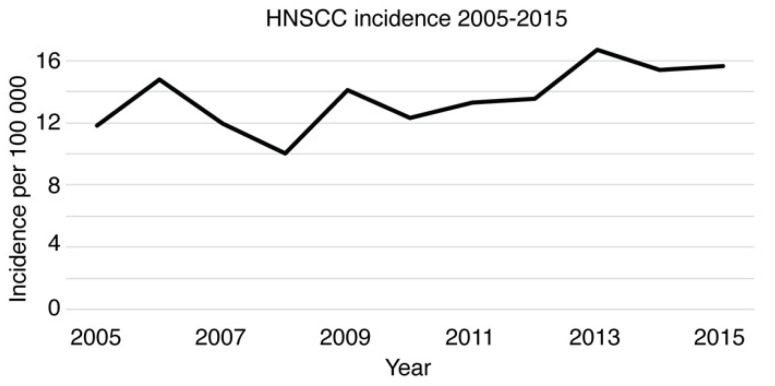
HNSCC incidence in the southwestern Finland population from 2005–2015.

**Figure 3 cancers-14-05717-f003:**
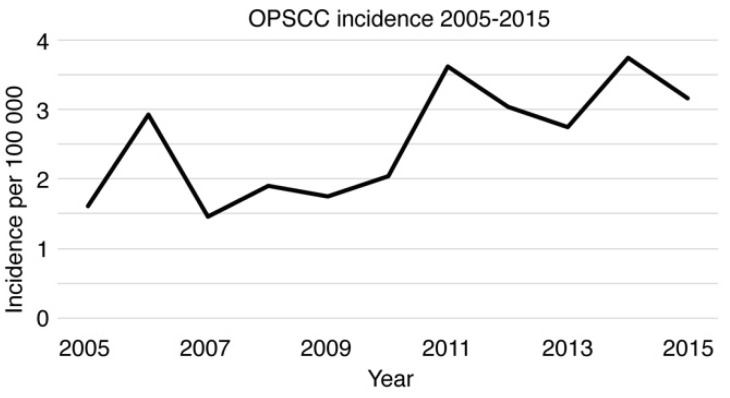
OPSCC incidence in the southwestern Finland population from 2005–2015.

**Figure 4 cancers-14-05717-f004:**
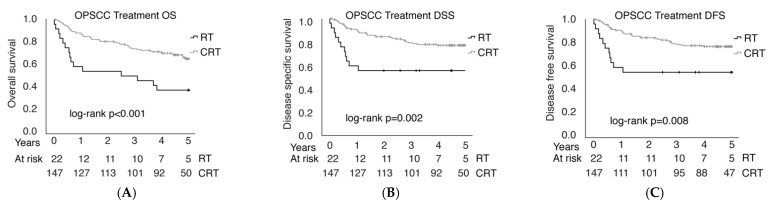
OPSCC overall survival (OS, **A**), disease-specific survival (DSS, **B**) and disease-free survival (DFS, **C**) after radio- (RT) or chemoradiotherapy (CRT).

**Figure 5 cancers-14-05717-f005:**
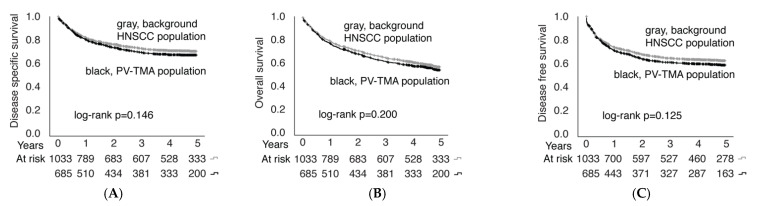
Overall survival (**A**), disease-specific survival (**B**) and disease-free survival (**C**) comparison between PV-TMA and background HNSCC populations.

**Figure 6 cancers-14-05717-f006:**
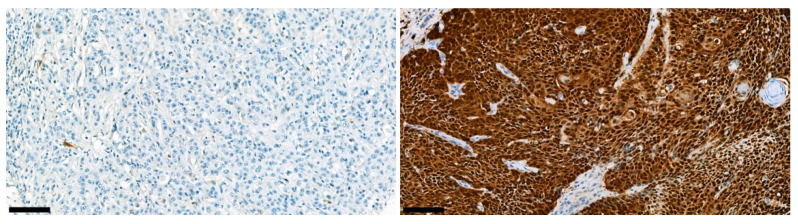
Representative immunohistochemical figures of negative (**left panel**) and positive (**right panel**) p16 staining. Black bars represent 100 µm.

**Figure 7 cancers-14-05717-f007:**
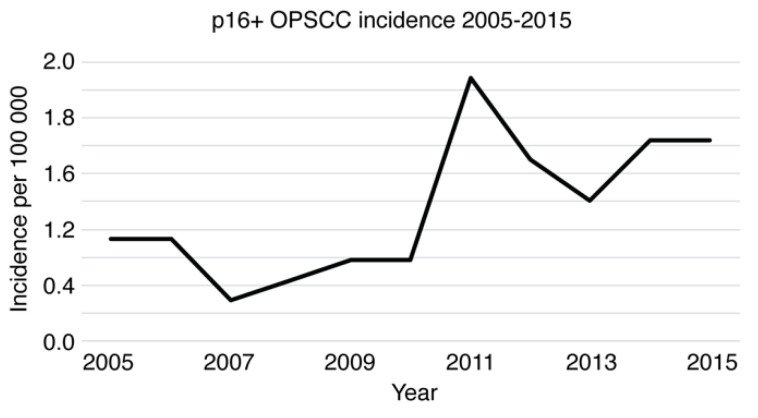
p16-positive OPSCC incidence in PV-TMA from 2005–2015.

**Figure 8 cancers-14-05717-f008:**
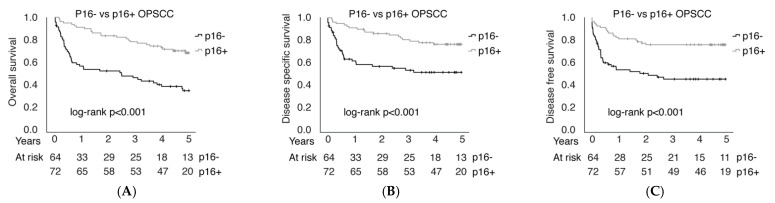
Overall survival (**A**), disease-specific survival (**B**) and disease-free survival (**C**) comparison between p16− and p16+ in OPSCC.

**Table 1 cancers-14-05717-t001:** Patient characteristics. RT (radiotherapy), CRT (chemoradiotherapy).

Characteristic	*n*	%	Survival Effect HR (95% CI)	*p* Value
**Gender**			Not included	
Male	679	65.7		
Female	354	34.3		
**Age**			1.03 (1.02–1.05)/year	<0.001
<65	487	47.1		
>65	546	52.9		
**Tumor site**			Not included	
Oral cavity	505	48.9		
Oropharynx	193	18.7		
Larynx	184	17.8		
Hypopharynx	40	3.9		
Other	111	10.8		
**T-class**				
T1-2	676	65.4	1	-
T3-4	357	34.6	2.45 (1.84–3.24)	<0.001
**N-class**				
N0	638	61.8	1	-
N+	395	38.2	1.79 (1.33–2.40)	<0.001
**Stage**			Not included	
0–II	481	46.6		
III–IV	552	53.4		
**Gradus**			Not included	
G1	321	32.9		
G2	435	44.6		
G3	219	22.5		
**Treatment**			Not included	
Surgery only	362	35.0		
Definitive RT	85	8.2		
Definitive CRT	159	15.4		
Surgery + CRT/RT	372	36.0		
Palliative treatment	55	5.3		
**Tobacco use**				
No	570	55.2	1	-
Yes	463	44.8	1.46 (1.07–1.99)	0.017
**Alcohol use**				
No	784	75.9	1	-
Yes	249	24.1	1.36 (1.00–1.85)	0.047

The survival effect was analyzed using a multivariable Cox proportional hazards model. The results include hazard ratios (HRs), 95% confidence intervals (CIs), and *p* values. Tobacco use was defined as daily smoking at the time of diagnosis. Alcohol use was defined as 10 doses or more a week at the time of diagnosis.

**Table 2 cancers-14-05717-t002:** Presence of known risk factors in different tumor sites of the head and neck region. Pack year (PY). Alcohol use was defined as 10 doses or more per week at the time of diagnosis.

	Smoking ≥20 PY (%)	Alcohol Current Use (%)	Gender Male (%)	Age ≥65 years (%)
Oral cavity (OSCC)	40.2	18.0	52.5	61.8
Oropharynx (OPSCC)	63.2	32.1	77.8	31.6
Larynx (LSCC)	83.2	32.1	86.4	48.9
Hypopharynx (HPSCC)	72.5	37.5	65.0	60.0
Other	38.7	19.8	71.2	53.2

**Table 3 cancers-14-05717-t003:** Univariate (left panel) and multivariate (right panel) analyses of TMA inclusion bias, including odds ratios (ORs), 95% confidence intervals (CIs), and *p* values. The results from binomial logistic modeling. Alcohol use was defined as 10 doses or more per week at the time of diagnosis.

	Total		TMA		TMA Inclusion		TMA Inclusion	
	*n*	%	*n*	%	OR (95% CI)	*p*	OR (95% CI)	*p*
**Gender**								
Male	679	66	438	64	0.79 (0.60–1.04)	0.089	0.77 (0.56–1.05)	0.103
Female	354	34	247	36	1	-	1	-
**Age**								
<65	487	47	334	49	1.21 (0.94–1.57)	0.145	1.02 (0.75–1.38)	0.904
≥65	546	53	351	51	1	-	1	-
**Smoker**								
<20 pack years	483	47	311	45	0.85 (0.66–1.10)	0.221	0.86 (0.62–1.18)	0.340
≥20 pack years	550	53	374	55	1	-	1	-
**Alcohol**								
No	784	76	513	75	0.85 (0.62–1.15)	0.290	0.97 (0.68–1.39)	0.877
Yes	249	24	172	25	1	-	1	-
**Tumor site**								
Oral cavity	505	49	352	51	1	-	1	-
Oropharynx	193	19	146	21	1.35 (0.92–1.97)	0.121	0.94 (0.60–1.49)	0.795
Larynx	184	18	109	16	0.63 (0.45–0.90)	0.010	0.69 (0.44–1.07)	0.097
Hypopharynx	40	4	30	4	1.30 (0.62–2.73)	0.482	0.92 (0.41–2.06)	0.836
Other	111	10	48	7	0.33 (0.22–0.50)	<0.001	0.23 (0.14–0.38)	<0.001
**T-class**								
T1–2	676	64	428	62	0.67 (0.51–0.89)	0.005	0.85 (0.51–1.39)	0.510
T3–4	357	36	257	38	1	-	1	-
**N-class**								
N0	638	62	387	57	0.50 (0.38–0.66)	<0.001	0.49 (0.30–0.81)	0.005
N+	395	38	298	43	1	-	1	-
**Stage**								
I–II	481	47	287	42	0.57 (0.44–0.74)	<0.001	0.94 (0.48–1.82)	0.853
III–IV	552	53	398	58	1	-	1	-
**Treatment**								
Surgery only	362	35	220	32	1	-	1	-
RT	85	8	45	7	0.73 (0.45–1.17)	0.187	0.66 (0.38–1.16)	0.146
CRT	159	15	108	16	1.37 (0.92–2.03)	0.120	1.25 (0.74–2.13)	0.408
RT + surgery	85	8	66	10	2.24 (1.29–3.90)	0.004	2.10 (1.17–3.78)	0.013
CRT + surgery	287	28	209	30	1.73 (1.24–2.42)	0.001	1.15 (0.73–1.38)	0.559
Palliative	55	5	37	5	1.33 (0.73–2.42)	0.357	0.98 (0.48–1.99)	0.951

**Table 4 cancers-14-05717-t004:** p16 staining results by primary tumor location. The “other” group consisted of tumors of the nasal cavity, nasopharynx, sinuses, and tumors of unknown primary location.

Site	p16+		p16−	
	*n*	%	*n*	%
OSCC	14	5	278	95
OPSCC	72	53	64	47
LSCC	5	5	89	95
HPSCC	1	4	24	96
Other	8	17	38	83

## Data Availability

The data presented in this study are available in this article.

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
