# Peer review of "Epidemiological Study of p16 Incidence in Head and Neck Squamous Cell Carcinoma 2005–2015 in a Representative Northern European Population"

_cancers, 2022, doi:10.3390/cancers14225717_

Round 1

Reviewer 1 Report

The paper is interesting as it witnesses that the epidemiologic development in Finland is similar of that described in many other countries. The strength is that the study can be regarded as population based because it includes all patients from the given region. Most of the similar data from other countries come from single institution studies only.

Material for p16 immunohistochemistry was available only for part (66%) of the cohort, regression analysis proved that the subgroup corresponds to the whole group.

In results authors state higher p16 positivity prevalence in oropharyngeal tumor than in tumors of other locations and the rising incidence of oropharyngeal tumors due to higher incidence of p16 positive tumors. All these are facts well known since many years from other countries, the paper documents that the development is similar in the given region. As in hundreds of other reports the better survival of p16 positive oropharyngeal tumors is also shown. Authors should adjust the survival on other prognostic factors at least on T and N classification and smoking. Authors also declare that the better survival is independent of treatment modality. This is highly probable, but the authors do not bring data supporting this. Survival in patients treated by different modalities is not shown. Authors also show that survival in patients treated by chemoradiation is better than in those treated by radiation. It should be clarified if also patients undergoing adjuvant radiation / chemoradiation are included in these curves. The group of patients treated by surgery only is probably not analyzed. This should be clarified. Furthermore such comparison without adjustment on at least T and N classification is problematic. Authors should complement it.

Author Response

Dear Reviewer,

We are grateful for the constructive remarks and comments by the reviewers on our paper. We have successfully addressed all the comments as described below point-by-point.

The paper is interesting as it witnesses that the epidemiologic development in Finland is similar of that described in many other countries. The strength is that the study can be regarded as population based because it includes all patients from the given region. Most of the similar data from other countries come from single institution studies only.

Material for p16 immunohistochemistry was available only for part (66%) of the cohort, regression analysis proved that the subgroup corresponds to the whole group.

Our reply:

We thank the reviewer for her/his thorough comments. We also fully agree with the reviewer that the observations made with a population-based approach and the verification of the similarity of the studied patient groups in multivariate analyzes are noteworthy in terms of the results and clinical applicability of our study.

In results authors state higher p16 positivity prevalence in oropharyngeal tumor than in tumors of other locations and the rising incidence of oropharyngeal tumors due to higher incidence of p16 positive tumors. All these are facts well known since many years from other countries, the paper documents that the development is similar in the given region. As in hundreds of other reports the better survival of p16 positive oropharyngeal tumors is also shown. Authors should adjust the survival on other prognostic factors at least on T and N classification and smoking. Authors also declare that the better survival is independent of treatment modality. This is highly probable, but the authors do not bring data supporting this. Survival in patients treated by different modalities is not shown. Authors also show that survival in patients treated by chemoradiation is better than in those treated by radiation. It should be clarified if also patients undergoing adjuvant radiation / chemoradiation are included in these curves. The group of patients treated by surgery only is probably not analyzed. This should be clarified. Furthermore such comparison without adjustment on at least T and N classification is problematic. Authors should complement it.

Our reply:

We thank the reviewer for her/his expert comments regarding the similarity and comparability of our results to earlier HNSCC, OPSCC and p16 publications. We have added the survival effect of T-class, nodal positivity, consumption of tobacco and alcohol, and age in Table1 and performed adjusted survival effects of p16 and treatment type. We clarified that also patients treated by adjuvant RT and CRT were included in the survival analyses and that the group of patients treated by surgery only was not analyzed.

In reply to this comment, we have added the following text in the Materials and Methods section: “The multivariable Cox proportional hazards method was applied to adjust the survival effect of p16 and treatment type on age, smoking, alcohol use, T-class, and nodal positivity. Hazard ratios (HR) with 95% confidence intervals (CI) and p values were reported."

Moreover, following text have been added in the revised manuscript to the end of the Results chapter: “To conclude, p16-positive patients had better OS, regardless of treatment modality (HR 0.64; 95% CI 0.43–0.95; p value 0.028). The survival effect was analyzed using a multivariable Cox proportional hazards model adjusting for T-class, nodal positivity, consumption of alcohol and tobacco, patient age, and treatment type.”

Reviewer 2 Report

Study evaluation

The authors have performed a reasonable epidemiological study of p16 incidence in head and neck squamous cell carcinoma by collecting all new patients in their targeted area over 10 years and correlating the gene expression to the clinical and epidemiological data that’s making their study a representative epidemiological study with originality to this research filed, but still, the patient's size is small to be considered an Epidemiological study.

Introduction

1.     There are missed links between the different paragraphs in the introduction section the state of art is missing and reorganizing the paragraphs will give more sense to the scientific story of the manuscript.

2.     This sentence must be revised to have more clarity to dusting between HPV- and HPV+ groups “Compared to HPV- negative OPSCC, patients with HPV-positive tumors tend to be younger and healthier and furthermore they typically smoke less, are males, have a higher average number of sexual partners and are geographically located in northern America and Europe [6,9-10]."

3.     The P16 paragraph must be revised and more details about P16 mechanisms in cancer in general and in HNSCC must be added to show the role of P16 in the increased frequency of HNSCC. 

4.     Authors must use scientific words that have been largely used in the file in the sentence below authors Wroten p16 has been accepted as a biomarker, we don’t use the word accepted it may be considered or approved… “Thus, p16 has been accepted as a biomarker for HPV-positive HNSCC 79 in clinical practise". 

Materials and Methods

1.     The quality of Figure 1 must be improved. 

2.     The overall survival was defined from end-of-treatment to end-of follow-up, authors must add a mean follow-up time by month for the whole patients.

3.     More details about the Immunohistochemistry protocol must be added. 

Results

1.     The quality of the Figures must be improved. 

2.     This paragraph “Locoregional epidemiology of HNSCC between the years 2005 – 2015” isn’t clear I can’t follow the author’s interpretation it must be rewritten.

3.     When authors compare two groups, they must mention both groups like in the sentence below larynx group was 42.1 % lower than all other groups or a specific group. “Meanwhile, the number of patients with a new SCC of the larynx 160 (LSCC) was 42.1 % lower”.

4.     We usually don’t use more strongly associated we use more promoted in males or just associated “Male gender was most 166 strongly associated with LSCC (86.4 %) and the OPSCC (77.8 %).”

Discussion

The discussion part must be more targeted to the study funding and the English must be improved. 

Author Response

Dear Reviewer,

We are grateful for the constructive remarks and comments by the reviewers on our paper. We have successfully addressed all the comments as described below point-by-point.

The authors have performed a reasonable epidemiological study of p16 incidence in head and neck squamous cell carcinoma by collecting all new patients in their targeted area over 10 years and correlating the gene expression to the clinical and epidemiological data that’s making their study a representative epidemiological study with originality to this research filed, but still, the patient's size is small to be considered an Epidemiological study.

Our reply:

We agree completely with the reviewer that the usage of the term "Epidemiology" should be precise and careful. We are also grateful for the reviewer's reasoned comment in this regard, as we have in fact spent considerable time ensuring that our study and HNSCC cohort meets strict epidemiological criteria.

The representativeness of population-based study design, the clinical HNSCC data, cancer sample collection and the fulfillment of our study in terms of epidemiological criteria have been already validated in our previous publication (PMID: 33582846). In this work Southwest Finland regional cohort corresponded with Finnish EUROCARE‐5 data (PMID: 26421817). In addition, in Finland all cancer patients, including patients suffering from HNSCC, are treated in five tertiary referral centers from which Southwest Finland tertiary referral center is one. In practice, HNSCC cancer treatment, including multi-disciplinary tumor board practice, meticulous treatment planning and impartial response monitoring are carried out in these centers in accordance with national recommendations. This type of centrally managed and implemented cancer treatment has ensured that cancer diagnosis, prevalence, follow-up and, above all, treatment results are very similar throughout Finland. For these reasons, we are fully convinced that the results of this population validated HNSCC study, in which all HNSCC patients from the Southwest Finland region have been collected and studied, can be scaled to the entire Finnish region in terms of results and findings. For these reasons, we consider that our study fully meets the criteria for using the term epidemiology.

In reply to this comment, we have added the following text in the introduction section: “The main purpose of this study was to examine the associations between p16 expression and overall survival (OS) among HNSCC patients in a representative and previously verified population-validated setting [32].”

Introduction

  1. There are missed links between the different paragraphs in the introduction section the state of art is missing and reorganizing the paragraphs will give more sense to the scientific story of the manuscript.

Our reply:

We agree with the reviewer that the introduction was not as clear as possible and in response, we have now clarified and unified the content and paragraphs of the introduction. In addition, the text of the introduction has been checked for language.

  1. This sentence must be revised to have more clarity to dusting between HPV- and HPV+ groups “Compared to HPV- negative OPSCC, patients with HPV-positive tumors tend to be younger and healthier and furthermore they typically smoke less, are males, have a higher average number of sexual partners and are geographically located in northern America and Europe [6,9-10]."

Our reply:

We thank the reviewer for this comment and agree that this sentence must be presented differently. In response, we have corrected this to following sentences: “Dividing OPSCC patients into two groups by HPV expression is useful, as HPV-positive and HPV-negative patients differ in characteristics: patients with HPV-positive tumors tend to be younger, healthier and smoke less. Furthermore, HPV-positive OPSCC patients are typically males and have a higher average number of sexual partners than HPV-negative OPSCC patients.”

  1. The P16 paragraph must be revised and more details about P16 mechanisms in cancer in general and in HNSCC must be added to show the role of P16 in the increased frequency of HNSCC.

Our reply:

We thank the reviewer for these comments and are in full agreement with reviewer that the p16 mechanisms and the role of p16 is important to show in the introduction. In reply to this comment, we have added the carcinogenesis of HPV infection and the role of p16 in HNSCC in the p16 paragraph.

  1. Authors must use scientific words that have been largely used in the file in the sentence below authors Wroten p16 has been accepted as a biomarker, we don’t use the word accepted it may be considered or approved… “Thus, p16 has been accepted as a biomarker for HPV-positive HNSCC 79 in clinical practise".

Our reply:

We are grateful for the reviewer to point this out and are in complete agreement that original sentence was not in accordance with scientific writing. In response, as we reorganized the introduction text and added more details about the p16 mechanism to the p16 paragraph, this sentence was excised.

Materials and Methods

  1. The quality of Figure 1 must be improved.

Our reply:

The quality of Figure 1 has been checked and improved. A new Figure 1, including better resolution, has been introduced in the revised manuscript.

  1. The overall survival was defined from end-of-treatment to end-of follow-up, authors must add a mean follow-up time by month for the whole patients.

Our reply:

We provided mean and median follow-up times by month in chapter 3.1 as requested. In reply to this comment, we have also added the following text in the Results chapter 3.1: “In addition, the survival effect of selected variables was calculated and reported based on a prognostic model previously described by Denissoff et al. [33]. The mean and median follow-up times were 38.6 and 49.3 months, respectively.”

  1. More details about the Immunohistochemistry protocol must be added.

Our reply:

A more detailed description of the implementation of immunohistochemistry, the antibody information and the performed analyzes have been added to the Immunohistochemistry section of Materials and Methods.

Results

  1. The quality of the Figures must be improved.

Our reply:

The qualities of Figures have been checked and improved. A new Figures, including better resolution, have been introduced in the revised manuscript.

  1. This paragraph “Locoregional epidemiology of HNSCC between the years 2005 – 2015” isn’t clear I can’t follow the author’s interpretation it must be rewritten.

Our reply:

We agree with the Reviewer that Results title 3.2. was not clear enough and was amended as follows: "3.2. Locoregional distribution of HNSCC between the years 2005 and 2015”.

  1. When authors compare two groups, they must mention both groups like in the sentence below larynx group was 42.1 % lower than all other groups or a specific group. “Meanwhile, the number of patients with a new SCC of the larynx 160 (LSCC) was 42.1 % lower”.

Our reply:

We thank the reviewer for her/his expert comments, based on which we have corrected this sentence according to reviewer’s comments.

  1. We usually don’t use more strongly associated we use more promoted in males or just associated “Male gender was most 166 strongly associated with LSCC (86.4 %) and the OPSCC (77.8 %).”

Our reply:

We thank the reviewer for these refinements, which we have paid attention to in our revised manuscript.

Discussion

The discussion part must be more targeted to the study funding and the English must be improved.

Our reply:

We thank the reviewer for these comments and have improved the content of the discussion paragraph. In addition, the English language of the entire manuscript has been revised and reviewed by a native English-speaking language reviewer.

Round 2

Reviewer 1 Report

Revised version along with a cover letter has sufficiently improved the manuscript to warrant publication in Cancers. 

Reviewer 2 Report

If authors improve the writing of the discussion manuscript can be accepted in cancers.